# Levosimendan Increases Survival in a D-Galactosamine and Lipopolysaccharide Rat Model

**DOI:** 10.3390/biomedicines10123161

**Published:** 2022-12-07

**Authors:** Tatsuma Sakaguchi, Fusao Sumiyama, Masaya Kotsuka, Masahiko Hatta, Terufumi Yoshida, Mikio Hayashi, Masaki Kaibori, Mitsugu Sekimoto

**Affiliations:** 1Department of Surgery, Kansai Medical University, Hirakata 573-1010, Japan; 2Department of Physiology, Kansai Medical University, Hirakata 573-1010, Japan

**Keywords:** sepsis, septic acute liver injury, levosimendan, D-galactosamine, lipopolysaccharide, inducible nitric oxide synthase, nuclear factor-κB

## Abstract

Levosimendan, a calcium sensitizer, has an organ protective profile through the inhibition of inflammatory mediators and cytokines in critical conditions, such as heart failure, ischemia-reperfusion injury, and sepsis. The survival effect of levosimendan for acute liver failure has not been examined yet. Male Sprague-Dawley rats were examined in the D-galactosamine hydrochloride and lipopolysaccharide (GalN/LPS) model. Levosimendan was injected intraperitoneally before GalN/LPS treatment. Survival was monitored for 7 days. For biochemical analyses, liver and blood samples were collected from the rats at 1 and 8 h after GaIN/LPS treatment. The pretreatment of levosimendan at 4 mg/kg significantly increased survival in GalN/LPS rats. In the liver specimen, levosimendan significantly inhibited the activation of nuclear factor-κB (NF-κB) at 1 h, and significantly decreased the mRNA expression of inflammatory mediators, including inducible nitric oxide synthase and tumor necrosis factor-α (TNF-α), at 8 h. In serum, levosimendan decreased the levels of nitrite, a metabolite of nitric oxide, and TNF-α protein, as well as aspartate aminotransferase and alanine aminotransferase. These results indicated that Levosimendan ameliorated liver dysfunction and survival in acute liver failure model rats through the suppression of NF-κB activation.

## 1. Introduction

Sepsis in liver failure or after hepatectomy remains a major issue. It is considered that endotoxin sensitivity is enhanced by the reduced phagocytic function of the reticuloendothelial system due to severe damage of the liver [1]. Endotoxins activate nuclear factor-κB (NF-κB) in macrophages, which has been assumed to be a trigger to induce expression of proinflammatory cytokines and inducible nitric oxide synthase (iNOS). The latter produces an excess of nitric oxide (NO), leading to septic shock and multiple organ failure [2]. However, there are no established pharmacologic interventions to prevent the cytokine storms. In practice, vasopressors are recommended as first-line agents for septic shock [3], but high doses of catecholamines and high levels of circulating catecholamines are associated with poor prognosis and severe side effects [4].

We previously reported that levosimendan, a calcium sensitizer, increased survival in lipopolysaccharide (LPS)-treated septic rats after 70% hepatectomy through mechanisms related to NO production [5]. Another in vivo model has also shown NO-mediated hepatoprotective effects of levosimendan in the ischemia-reperfusion model [6]. Levosimendan downregulates NO production in response to inflammatory stimuli, which is probably mediated through different signaling pathways among cell types [6,7,8]. It is widely recognized that NF-κB plays an important role as a transcriptional factor of the iNOS gene, which results in iNOS and NO production. However, levosimendan did not modify the DNA binding of NF-κB in response to LPS in J774 macrophages in vitro [8]. In addition, levosimendan did not show a significant effect on NF-κB activation in hepatectomy with LPS model rats [5]. Thus, the mechanism of the effect by levosimendan has been unclear.

The aim of the present study was to clarify whether levosimendan improves survival in another experimental model of sepsis with acute liver injury, that is, simultaneous administration of D-galactosamine hydrochloride and lipopolysaccharide (GalN/LPS) in rats. GalN/LPS coinjection is a common method to establish an acute liver injury in rats, which increases the sensitivity to LPS-induced hepatotoxicity [9]. Furthermore, we demonstrated that levosimendan reduced the inflammatory mediators in the liver and serum of the sepsis rats through NF-κB suppression.

## 2. Materials and Methods

### 2.1. Ethics

Animal care and experiments were performed in accordance with the standards in the ARRIVE and PREPARE guidelines [10,11]. In addition to these, our study was in accordance with the relevant guidelines and regulations approved by the Animal Care Committee of Kansai Medical University (approval number 18-027(01) of 4 April 2018). All methods proposed in these studies were also carried out according to the standards of relevant institutional guidelines and regulations.

### 2.2. Drugs

Levosimendan was purchased from Wako Pure Chemical Industries (Osaka, Japan). Levosimendan was resolved in dimethyl sulfoxide (DMSO) and stored at −80 °C. For GalN/LPS experiments, resolved levosimendan was diluted by 1 mL normal saline for each rat that the concentration of DMSO was decided at 2%. Recombinant human interleukin-1β (IL-1β; 2 × 10^7^ U/mg protein) was purchased from MyBioSource (San Diego, CA, USA). Isoflurane, pentobarbital sodium, collagenase, a Transaminase CII-test kit, GalN, 10% formalin, and a PicaGene Luminescence kit were obtained from Wako Pure Chemical Industries (Osaka, Japan). LPS (Escherichia coli; O111:B4) and mouse anti-β-tubulin were obtained from Sigma–Aldrich (St. Louis, MO, USA). Enzyme-linked immunosorbent assay (ELISA) kits were obtained from Life Technologies (Carlsbad, CA, USA). TRIzol Reagent was obtained from Thermo Scientific (Waltham, MA, USA). T4 polynucleotide kinase, Oligo (dT) Primer, dNTPs Mixture, RNase Inhibitor, and Rever Tra Ace were obtained from Toyobo (Osaka, Japan). Beta-Glo kits and mouse immunoglobulin κ light chain were obtained from Promega (Madison, WI, USA).

### 2.3. Animals

Male Sprague-Dawley rats [specific pathogen free, 8 weeks old, 280–320 g] were purchased from Charles River Laboratories Japan (Yokohama, Japan) and housed in the animal care facility of the Laboratory Animal Center, Kansai Medical University, Japan, under a 12-h light–dark cycle (lights on at 8:00). Constant temperature (21–23 °C) and relative humidity (40–60%) were maintained. The animals had free access to a diet of γ-irradiated CRF-1 (Oriental Bioservice, Kyoto, Japan) and water. Every effort was made to reduce the number of the animals used and minimize animal suffering.

### 2.4. D-Galactosamine Hydrochloride and Lipopolysaccharide (GalN/LPS) Model

Acute liver injury was induced in the in vivo model. Male Sprague-Dawley rats were anesthetized with isoflurane (Abbott Laboratories, Abbott Park, IL, USA) before receiving the treatment. The rats were intraperitoneally injected with levosimendan at 2 or 4 mg/kg or saline (positive control; PC). One hour later, a mixture of 500 mg/kg GalN and 2.5 µg/kg LPS (GalN/LPS) was injected via the penile vein [12]. Survival was monitored in 12 rats in each group for 7 days. The rats were killed when they appeared weak and moribund because of the progression of liver failure, congestion, and multi-organ failure. We used the NIH Office of Animal Care and Use score to assess severity before liver resection [13]. As a separate experiment from the survival analysis, liver and blood samples were collected at both 1 and 8 h after GaIN/LPS treatment and stored at −80 °C. Sample size was determined as follows: levosimendan (4 mg/kg) group (*n* = 6), PC group (*n* = 4), and negative control group (*n* = 1). In order to detect early and late responses of mediators during inflammation, blood and liver samples were collected at 1 and 8 h.

### 2.5. Electrophoretic Mobility Shift Assay (EMSA)

EMSA was performed as described previously with a minor modification [14]. Nuclear extracts were prepared from frozen liver. Binding reactions were undertaken by incubating the nuclear extracts (4 μg) in reaction buffer (20 mM HEPES-KOH, pH 7.9, containing 1 mM EDTA, 60 mM KCl, 10% glycerol, and 1 µg poly[dI-dC]) with a probe (40,000 dpm)) for 20 min at room temperature. Products were electrophoresed on a 4.8% polyacrylamide gel in high-ionic-strength buffer, and dried gels were analyzed by autoradiography. An NF-κB consensus oligonucleotide (5′-AGTTGAGGGGACTTTCCCAGGC) from the mouse immunoglobulin κ light chain was labelled with [γ-^32^P]-ATP (PerkinElmer, Tokyo, Japan) and T4 polynucleotide kinase. Protein was measured using the Bradford method [15]. Bands corresponding to NF-κB were quantified by densitometry using ImageJ [16].

### 2.6. Reverse Transcriptase-Polymerase Chain Reaction (RT-PCR)

Total RNA was extracted from the frozen liver samples using TRIzol reagent [17]. cDNA was synthesized from 1 µg total RNA using Rever Tra Ace with oligo(dT)20 primer. The conditions of thermal cycling using iCycler (Bio-Rad Laboratories, Hercules, CA, USA) were 42 °C for 60 min and 95 °C for 5 min. Real-time PCR was performed using SYBR Green and primers for each gene. Primer sequences were synthesized by Eurofins Genomics (Tokyo, Japan) (Table 1). The conditions of thermal cycling using a Rotor-Gene Q (Qiagen, Stanford, VA, USA) were 95 °C for 5 min followed by 40 cycles of 95 °C for 5 s and 60 °C for 10 s. Collection and analyses of data were undertaken using the system software. mRNA expression levels of each gene were measured as CT threshold levels and normalized to those of eukaryotic elongation factor-1α.

### 2.7. Serum Biochemical Analysis

Serum alanine aminotransferase (ALT) and aspartate aminotransferase (AST) levels were quantified using commercial kits. The serum levels of nitrite and nitrate, stable metabolites of NO, were measured using a commercial kit (Roche, Mannheim, Germany) according to the Griess method [18]. Serum levels of tumor necrosis factor-α (TNF-α), interleukin-6 (IL-6), and IL-1β were determined using commercial kits (Thermo Fisher Scientific, Waltham, MA, USA).

### 2.8. Statistical Analysis

Quantitative results were obtained from three to four independent experiments for each of the various analyses, and the mean values and their standard deviations (SD) were calculated. Differences between groups and survival rates were analyzed by the Student’s *t*-test and log-rank test, respectively. The Bonferroni method was used as multiple tests for comparison of survival curves (JMP 16; SAS Institute, Cary, NC, USA). *p* < 0.05 was considered significant.

## 3. Results

### 3.1. Levosimendan Increases Survival in GalN/LPS Rats

Rats were treated with levosimendan (2 and 4 mg/kg, i.p.) or saline (PC) at 1 h before GalN/LPS injection. All rats that survived at 48 h survived until 7 days later (Figure 1). The survival rate in the PC group at day 1 and 2 was 13% and 6.3%, respectively. The survival rate in the 2 mg/kg levosimendan pretreatment group at day 1 and 2 was 44% and 19%, respectively. In the pretreatment group of levosimendan at 4 mg/kg, survival significantly increased (*p* < 0.01), and the survival rate on day 1 and 2 was 82% and 63%, respectively.

### 3.2. Levosimendan Inhibits the Activation of NF-κB in the Liver of GalN/LPS Rats

The levosimendan pretreatment (4 mg/kg) significantly decreased the activation of NF-κB at 1 h (*p* = 0.04, Figure 2), whereas there was no difference at 8 h.

### 3.3. Levosimendan Decreases the Inflammatory Mediator but Increases IL-10 mRNA in the Liver of GalN/LPS Rats

The pretreatment group of levosimendan (4 mg/kg) significantly decreased the levels of mRNA of iNOS, TNF-α, IL-1β, cytokine-induced neutrophil chemoattractant-1 (CINC-1), and IL-6 compare to those of the PC group at 8 h (*p* = 0.02, 0.02, 0.03, 0.04, and < 0.01, respectively, Figure 3). In addition, levosimendan significantly decreased the levels of IL-1β mRNA at 1 h (*p* = 0.02). In contrast, levosimendan significantly increased interleukin-10 (IL-10) mRNA at 1 h (*p* = 0.03).

### 3.4. Levosimendan Decreases NO, AST/ALT, and Cytokines in Serum of GalN/LPS Rats

The serum levels of NO and liver enzymes (AST and ALT) increased 8 h after GalN/LPS injection, suggesting acute liver injury (Figure 4). In addition, IL-6 and IL-1β were elevated in the serum after GalN/LPS injection. The pretreatment group of levosimendan (4 mg/kg) significantly decreased the levels of NO, AST, and ALT at 8 h (*p* < 0.01, 0.03, and 0.01, respectively). Additionally, levosimendan significantly decreased TNF-α at 1 h (*p* = 0.01) and IL-6 and IL-1β at 8 h (*p* < 0.01).

## 4. Discussion

In the present study, we demonstrated that the pretreatment of levosimendan improved the survival of an experimental model of sepsis with acute liver injury (GalN/LPS model rats) in a dose-dependent manner (Figure 1) and suppressed NF-κB activation in the liver (Figure 2). Levosimendan decreases the mRNA of the inflammatory mediators, such as iNOS, TNF-α, CINC-1, and IL-6, in the liver of GalN/LPS rats (Figure 3). Furthermore, the levosimendan pretreatment reduced NO, TNF-α, IL-6, and IL-1β, as well as liver enzymes, in the serum of the sepsis rats, suggesting a hepatoprotective effect (Figure 4).

Levosimendan inhibits a transcriptive factor of hypoxia-inducible factor-1 (HIF-1) expression [19]. The activation of HIF-1 suppressed IL-1β-induced iNOS mRNA expression and subsequent NO production in primary cultured rat hepatocytes [20]. In this study, the pretreatment of levosimendan significantly decreased the activation of NF-κB at 1 h of GalN/LPS treatment, but not at 8 h. The inhibitory effect of NF-κB at the early phase is essential [21]. NF-κB translocates into nuclei and binds to B motif in the promoters of pro-inflammatory genes, such as iNOS, TNF-α, and IL-1β, leading to the induction of their mRNA expression [22]. In contrast, mRNA of IL-10, which is a pleiotropic cytokine known for its potent anti-inflammatory and imuno-suppressive effects [23], was enhanced in the liver of levosimendan-pretreated rats. In a previous study, levosimendan did not suppress NF-κB activation in 70% hepatectomy with the LPS model [5]. The discrepancy indicates that levosimendan activates different signaling pathways in hepatocytes than in other cell types, such as macrophages. LPS directly affects macrophages to activate NF-κB, which is a transcriptive factor involved in inflammatory cytokines [24]. In this study, significant differences in NO, AST, ALT, IL-6, and IL-10 were detected in blood samples at 8 h, but not at 1 h.

Growing evidence has shown the benefit of levosimendan on cardiocerebral, cardiopulmonary, cardiohepatic, and cardiorenal syndromes. The clinical use of levosimendan in cardiogenic shock during sepsis is still under debate [25]. Levosimendan compared to dobutamine showed more vasodilation and less inotropic activity in patients undergoing mitral valve surgery for mitral stenosis [26]. For sepsis-induced cardiac dysfunction, levosimendan may have benefits embodied in cardiac function improvement, though the effect of levosimendan was not superior to dobutamine [27]. We demonstrated an inhibitory effect on inflammatory cytokines and liver protection in rats, which may contribute to organ protection in sepsis, suggesting increased survival in cases of sepsis after liver failure (Figure 1). As there is some clinical evidence that levosimendan has beneficial potential on renal [28], pulmonary [29], and hepatic [30] function in patients with sepsis, future studies of levosimendan should focus on organ protection through NF-κB and NO suppression. On human hepatocytes, levosimendan showed the protective effect in ischemia-reperfusion injury through preventing down-regulation of the anti-apoptotic protein Bcl-2 (B-cell lymphoma 2), as well as up-regulation of the pro-apoptotic protein BAX (BCL2 associated X) [31].

Several limitations of this study should be acknowledged. We used intraperitoneal bolus administration of levosimendan before one hour of GalN/LPS administration. Correspondingly, previous reports showed that posttreatment of levosimendan had no effect on survival in 70% hepatectomy with the LPS model [5]. One may argue that intraperitoneal bolus administration does not represent the clinical situation. Although the half-life of levosimendan is about 1 h in humans, its active metabolite, OR-1896, has a half-life of 80 h [32], which could cover the duration of the effect by GalN/LPS administration. GalN depletes UTP and inhibits RNA synthesis in hepatocytes. This metabolic arrest increases sensitivity to the lethal effects of LPS administration in the liver [9]. However, we cannot rule out the contribution of other organs, such as the kidneys, lungs, and cardiovascular system. We did not include a sham surgical control because creating a GalN/LPS rat model does not require invasive surgical techniques. Future studies are needed to establish the mechanism of effect on acute liver injury using an ex vivo model.

In conclusion, the pretreatment of levosimendan improved survival in GalN/LPS sepsis model rats. Levosimendan reduced the inflammatory mediators through suppression of NF-κB activation in the liver. Levosimendan would have a beneficial effect in liver surgery and transplantation, as well as sepsis management after acute liver injury.

## Figures and Tables

**Figure 1 biomedicines-10-03161-f001:**
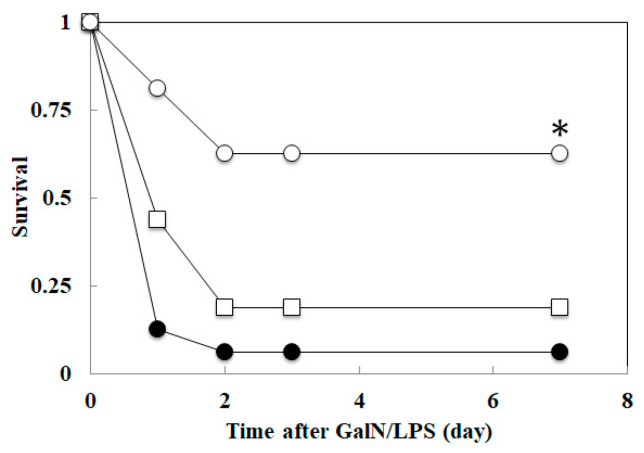
Effect of levosimendan on survival in D-galactosamine hydrochloride and lipopolysaccharide (GalN/LPS) rats. A mixture of 500 mg/kg GalN and 2.5 µg/kg LPS (GalN/LPS) was injected via the penile vein. The curves are plotted survival rate at 1, 2, 3, and 7 days. The survival rate in the positive control group (filled circle), the pretreatment group of levosimendan 2 mg/kg (square), or 4 mg/kg (open circle). * *p* < 0.05 compared with the positive control group by log-rank test.

**Figure 2 biomedicines-10-03161-f002:**
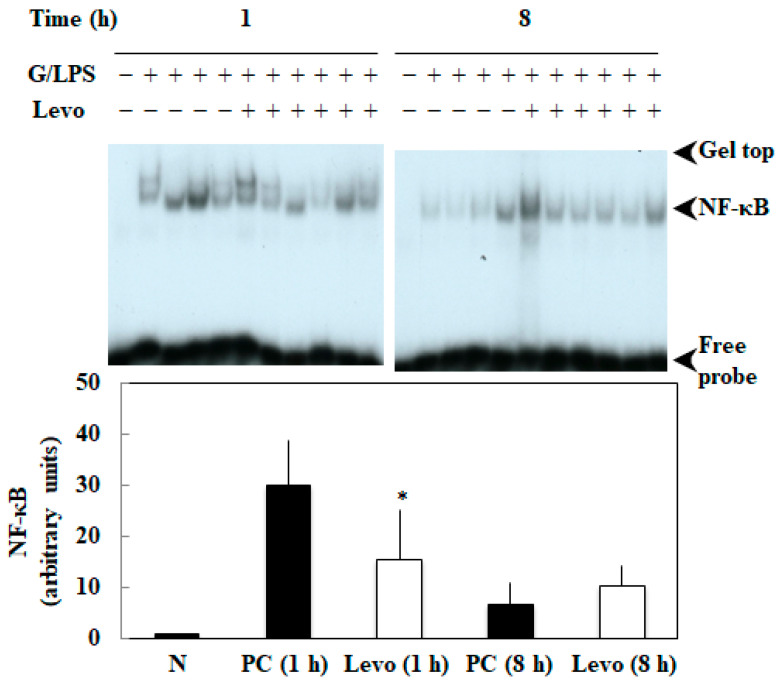
Effect of levosimendan on activation of NF-κB in the livers of D-galactosamine hydrochloride and lipopolysaccharide (GalN/LPS) rats. A mixture of 500 mg/kg GalN and 2.5 µg/kg LPS (G/LPS) was injected. The GalN/LPS rats were treated with levosimendan at 4 mg/kg (Levo). Upper, activation of NF-κB was examined by electrophoretic mobility shift assay. Lower, the bands corresponding to NF-κB were quantitated by densitometry. N, normal rat; PC, positive control. “+” indicates that G/LPS or Levo was administered; “−” indicates that GalN/LPS or Levo was not administered. * *p* < 0.05 compared with PC.

**Figure 3 biomedicines-10-03161-f003:**
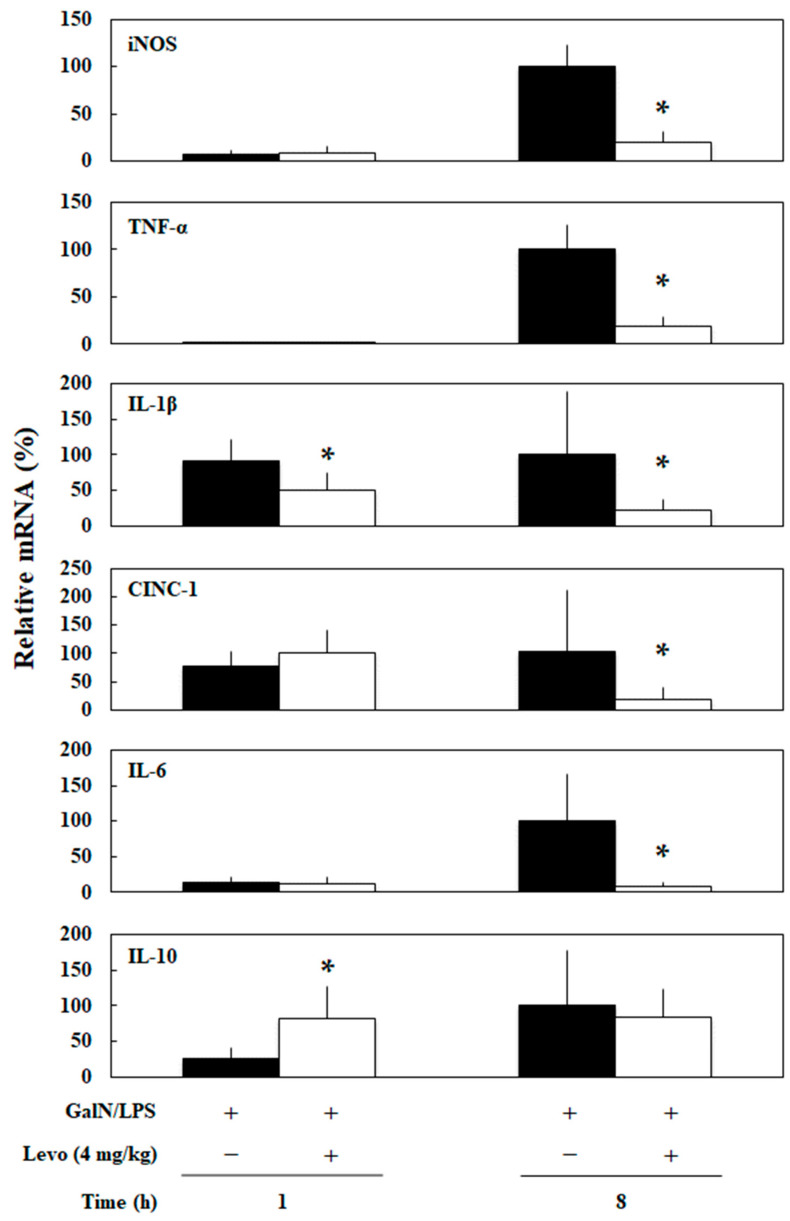
Effect of levosimendan (Levo) on mRNA expression of inflammatory mediators in the livers of D-galactosamine hydrochloride and lipopolysaccharide (GalN/LPS) rats. Each graph consists of bars representing pretreatment with levosimendan or vehicle 1 or 8 h after GalN/LPS injection (*n* = 5). “+” indicates that GalN/LPS or Levo was administered; “−” indicates that GalN/LPS or Levo was not administered. * *p* < 0.05 compared with vehicle control. iNOS, inducible nitric oxide synthase; TNF-α, tumor necrosis factor-α; IL-1β, interleukin-1β; CINC-1, cytokine-induced neutrophil chemoattractant-1; IL-6, interleukin-6; IL-10, interleukin-10.

**Figure 4 biomedicines-10-03161-f004:**
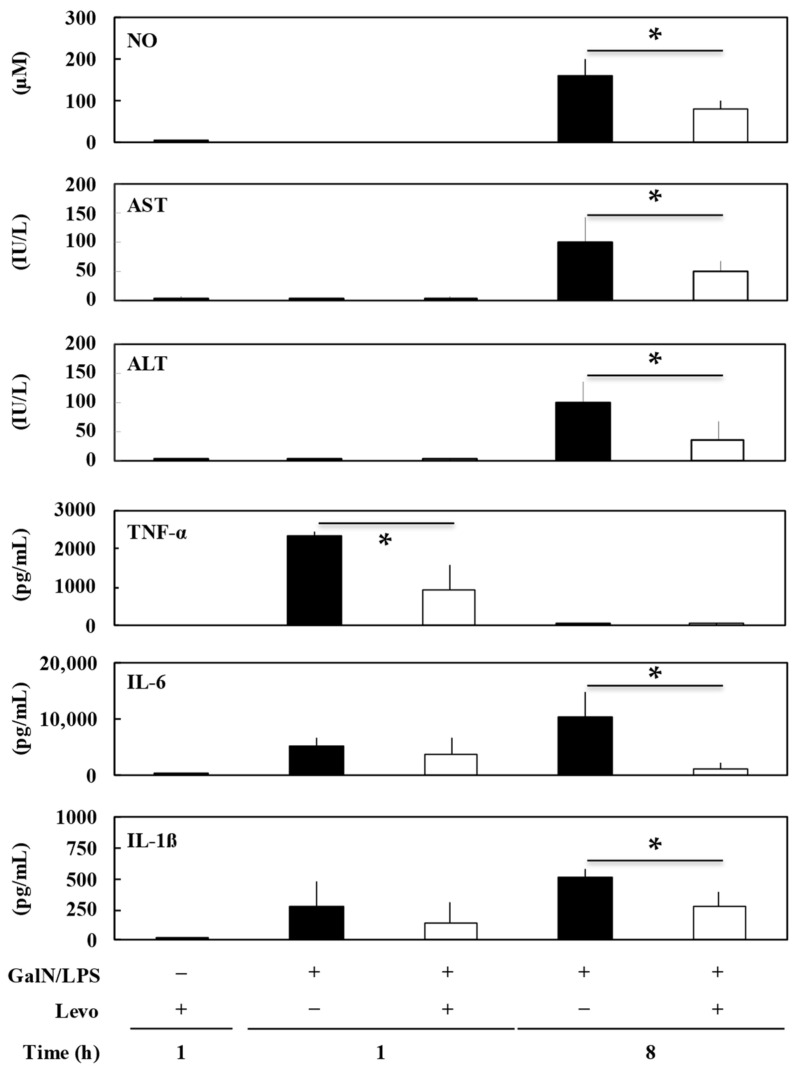
Effect of levosimendan (Levo) on serum level of inflammatory mediators in the livers of D-galactosamine hydrochloride and lipopolysaccharide (GalN/LPS) rats. Each graph consists of bars representing pretreatment with levosimendan or vehicle after GalN/LPS injection (*n* = 5). “+” indicates that GalN/LPS or Levo was administered; “−” indicates that GalN/LPS or Levo was not administered. * *p* < 0.05 compared with vehicle control. NO, nitric oxide; AST, aspartate transaminase; ALT, alanine transaminase; TNF-α, tumor necrosis factor-α; IL-6, interleukin-6; IL-1β, interleukin-1β.

**Table 1 biomedicines-10-03161-t001:** List of primer sets for real-time PCR.

Gene	Forward	Reverse
iNOS	5′-CCAACCTGCAGGTCTTCGATG-3′	5′-GTCGATGCACAACTGGGTGAAC-3′
TNF-α	5′-TCCCAACAAGGAGGAGAAGTTCC-3′	5′-GGCAGCCTTGTCCCTTGAAGAGA-3′
IL-1β	5′-TCTTTGAAGAAGAGCCCGTCCTC-3′	5′-GGATCCACACTCTCCAGCTGCA-3′
CINC-1	5′-GCCAAGCCACAGGGGCGCCCGT-3′	5′-ACTTGGGGACACCCTTTAGCATC-3′
IL-6	5′-GAGAAAAGAGTTGTGCAATGGCA-3′	5′-TGAGTCTTTTATCTCTTGTTTGAAG-3′
IL-10	5′-GCAGGACTTTAAGGGTTACTTGG-3′	5′-CCTTTGTCTTGGAGCTTATTAAA-3′
EF	5′-TCTGGTTGGAATGGTGACAACATGC-3′	5′-CCAGGAAGAGCTTCACTCAAAGCTT-3′

iNOS, inducible nitric oxide synthase; TNF-α, tumor necrosis factor-α; IL-1β, interleukin-1β; CINC-1, cytokine-induced neutrophil chemoattractant-1; IL-6, interleukin-6; IL-10, interleukin-10; EF, elongation factor-1α.

## Data Availability

The data that support the findings of this study are available on request from the corresponding author.

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
