# Peer review of "Levosimendan Increases Survival in a D-Galactosamine and Lipopolysaccharide Rat Model"

_biomedicines, 2022, doi:10.3390/biomedicines10123161_

Round 1

Reviewer 1 Report

An interesting in vivo study concerning the effect of levosimendan on acute liver failure.

However, I would like to mention the following points:

- In Figure 1, the authors should add the SD.

I understand that each group had 12 animals monitored for survival and the studies concerning blood and liver samples were conducted in 5 animals. Why did the authors select 5 animals and were these animals  selected randomly?

-I would like to see a comment concerning the pre-treatment with levosimendan ( the selection of 1 hour prior the  G/LPS administration), as well as the selection of the 1h and 8h time-points after.

-I have some concerns about the electrophoresis results presented in Figure 2. The blots are not clear and so I am not sure about the equal protein loading based on the β-tubulin.  I believe that the authors should replace the blots by better images, as N=4.

Author Response

We want to thank Reviewer #1 for providing valuable comments that helped improve the quality of the manuscript.  As shown in the responses, all comments were taken into consideration.

Comments from reviewer #1:

  1. In Figure 1, the authors should add the SD.

Based on your suggestion, the following sentence was added (line 154):  “The curves are plotted survival rate at 1, 2, 3, and 7 days.”

  1. I understand that each group had 12 animals monitored for survival and the studies concerning blood and liver samples were conducted in 5 animals. Why did the authors select 5 animals and were these animals selected randomly?

Based on your suggestion, the following sentences were added (lines 95–99):  “As a separate experiment from survival analysis, liver and blood samples were collected at both 1 and 8 hours after GaIN/LPS treatment and stored at −80 °C. Sample size was determined as follows: levosimendan (4 mg/kg) group (n = 6), PC group (n = 4), and negative control group (n = 1).”

  1. I would like to see a comment concerning the pre-treatment with levosimendan ( the selection of 1 hour prior the G/LPS administration), as well as the selection of the 1h and 8h time-points after.

Based on your suggestion, the following sentences were added (lines 235–238):  “We used intraperitoneal bolus administration of levosimendan before one hour of GalN/LPS administration. Correspondingly, previous report showed that posttreatment of levosimendan had no effect on survival in 70% hepatectomy with LPS model [5].”

Lines 99–100:  “In order to detect early and late responses of mediators during inflammation, blood and liver samples were collected at 1 and 8 hours.”

Lines 216–218:  “In this study, significant differences in NO, AST, ALT, IL-6, and IL-10 were detected at 8 hours, but not at 1 hour in blood samples.

  1. I have some concerns about the electrophoresis results presented in Figure 2. The blots are not clear and so I am not sure about the equal protein loading based on the β-tubulin. I believe that the authors should replace the blots by better images, as N=4.

Based on your suggestion, Figure 2A is now withdrawn because there is no clear image to replace it.

We want to thank you for providing positive and helpful comments, and we hope that the manuscript is now acceptable for publication.

Reviewer 2 Report

Well written important topic 

line 176 should this read mRNA of inflammatory mediators?

213 should change treatment with pretreatment 

218 Consider modifying sentence to that the data supports that levosimendan increase IL-10 transcription.

Inquiry: Why was levosimendan administered before inducing sepsis?

I understand that no one wants to waste animal resources have any of your prior studies have a sham surgical control? should consider commenting this in discussion on limitations  

Author Response

We want to thank reviewer #2 for providing valuable comments that helped improve the quality of the revised manuscript.  As shown in the responses, all comments were considered.

Comments from reviewer #2:

Comment 1:

  1. line 176 should this read mRNA of inflammatory mediators?

Based on your suggestion, the following sentence was revised (lines 169–172):  “The pretreatment group of levosimendan (4 mg/kg) significantly decreased the levels of mRNA of iNOS, TNF-α, IL-1β, cytokine-induced neutrophil chemoattract-ant-1 (CINC-1), and IL-6 compare to those of the PC group at 8 hours (p = 0.02, 0.02, 0.03, 0.04, and < 0.01, respectively, Figure 3).”

  1. 213 should change treatment with pretreatment

The “treatment” of levosimendan was replaced with “pretreatment” throughout the revision.

3.   218 Consider modifying sentence to that the data supports that levosimendan increase IL-10 transcription.

Based on your suggestion, the following sentence was revised (lines 210–212):  “In contrast, mRNA of IL-10, which is a pleiotropic cytokine known for its potent anti-inflammatory and imuno-suppressive effects [23], was enhanced in the liver of levo-simendan pretreated rats.”

4.Inquiry: Why was levosimendan administered before inducing sepsis?

Based on your suggestion, the following sentences were added (lines 235–238):  “We used intraperitoneal bolus administration of levosimendan before one hour of GalN/LPS administration. Correspondingly, previous report showed that posttreatment of levosimendan had no effect on survival in 70% hepatectomy with LPS model [5].”

  1. I understand that no one wants to waste animal resources have any of your prior studies have a sham surgical control? should consider commenting this in discussion on limitations

Lines 245–246:  “We did not include a sham surgical control because it does not require invasive surgical technique to create GalN/LPS rat model.”

We want to thank you for providing positive and helpful comments, and we hope that the manuscript is now acceptable for publication.

Round 2

Reviewer 1 Report

Τhe authors made the corrections and answered my queries.